# Total Soluble Solids in Grape Must Estimation Using VIS-NIR-SWIR Reflectance Measured in Fresh Berries

**Karen Brigitte Mejía-Correal** [1,*][ID], **Víctor Marcelo** [2][ID], **Enoc Sanz-Ablanedo** [1][ID]
and **José Ramón Rodríguez-Pérez** [1][ID]

1   Grupo de Investigación en Geomática e Ingeniería Cartográfica (GEOINCA), Universidad de León, Avenida de Astorga sn, 24401 Ponferrada, León, Spain; esana@unileon.es (E.S.-A.); jr.rodriguez@unileon.es (J.R.R.-P.)
2   Departamento de Ingeniería y Ciencias Agrarias, Universidad de León, Avenida de Astorga sn, 24401 Ponferrada, León, Spain; v.marcelo@unileon.es
*   Correspondence: kmejc@unileon.es

**Abstract:** Total soluble solids (TSS) is a key variable taken into account in determining optimal grape maturity for harvest. In this work, partial least square (PLS) regression models were developed to estimate TSS content for Godello, Verdejo (white), Mencía, and Tempranillo (red) grape varieties based on diffuse spectroscopy measurements. To identify the most suitable spectral range for TSS prediction, the regression models were calibrated for four datasets that included the following spectral ranges: 400–700 nm (visible), 701–1000 nm (near infrared), 1001–2500 nm (short wave infrared) and 400–2500 nm (the entire spectral range). We also tested the standard normal variate transformation technique. Leave-one-out cross-validation was implemented to evaluate the regression models, using the root mean square error (RMSE), coefficient of determination ($R^2$), ratio of performance to deviation (RPD), and the number of factors (F) as evaluation metrics. The regression models for the red varieties were generally more accurate than the models of those for the white varieties. The best regression model was obtained for Mencía (red): $R^2$ = 0.72, RMSE = 0.55 °Brix, RPD = 1.87, and factors n = 7. For white grapes, the best result was achieved for Godello: $R^2$ = 0.75, RMSE = 0.98 °Brix, RPD = 1.97, and factors n = 7. The methodology used and the results obtained show that it is possible to estimate TSS content in grapes using diffuse spectroscopy and regression models that use reflectance values as predictor variables. Spectroscopy is a non-invasive and efficient technique for determining optimal grape maturity for harvest.

**Keywords:** VIS-NIR spectroscopy; PLS regression; viticulture; total soluble solids

## 1. Introduction

Given that viticulture is a fundamental sector in the Spanish autonomous community of Castilla y León (some 70,000 planted hectares, 60% registered under the designation of origin (DO) label), obtaining high-quality wines relies on grape harvesting at the optimum point of ripeness, determined by grape content in total soluble solids (TSS) [1]. Ripeness is related to must sugar content, which is usually estimated from the refractive index for the sample as TSS content [2]. While many variables play a role in optimal maturity, including acidity, weight, anthocyanin content, and phenol maturity, TSS is the fundamental parameter used to decide the harvest date [3].

Since TSS in fruit juices, as measured by refractometry, is destructive, time-consuming, and environmentally damaging, today's viticulture requires property evaluation in a more timely and cost-effective way [4]. Near-infrared (NIR) spectroscopy is a feasible approach to measuring compounds in organic and biological substances, including grape samples [5], given that it is a powerful and non-invasive technique. With the development of cheaper, faster, and more accurate tools and sensors, NIR is increasingly being applied in the food industry [4,5], and likewise, it has become a very popular technique for the non-invasive assessment of intact fruit [6,7]. Spectroscopy has been used to analyze parameters

such as TSS, pH, and titratable acidity (TA) in juices and intact fruits in the range of 380–2500 nm [8,9].

One recent study obtained reflectance spectra from grapes in the field using a portable contact probe spectrometer [10] and predicting TSS from visible (VIS) to NIR and shortwave infrared (SWIR) hyperspectral data applying four different machine learning algorithms (partial least squares regression (PLS), random forest regression, support vector regression, and convolutional neural networks (CNN)). To evaluate the model's performance more robustly, 5-fold cross-validation (CV) was employed based on three metrics: the coefficient of determination ($R^2$), root mean square error (RMSE), and the ratio of performance to interquartile distance (RPIQ). The best models were obtained using the CNN learning algorithm ($R^2 = 0.63$, RMSE = 2.10, and RPIQ = 2.24 for the Chardonnay variety).

Another study has measured reflectance spectra from grape berry bunches in the laboratory using Fourier-transform long-wavelength near-infrared (NIR) reflectance spectra to predict TSS [11]. The dataset was divided into training and test sets for modeling: 80% for training and 20% for validation using multiple linear regression (MLR), and 90% for training and 10% for CV validation in 10 rounds using PLS methods. Variable selection further improved the prediction accuracy for both approaches ($R^2 = 0.972$ and 0.926, and RMSE = 0.306 and 0.472 for MLR and PLS, respectively).

In this research, spectral measurements were obtained from grape samples of 100 intact berries using a pistol grip in laboratory conditions. These grape samples were collected from four different grape varieties that were compared. While several authors have performed measurements in different vegetative stages [4,8,10,11], we collected the grape samples one day before harvest to reproduce an actual scenario and to take advantage of the sampling that winegrowers typically carry out before the harvest. Our aim was to develop a method that reduced variability and used fewer samples, a pistol grip to replicate field conditions, and obtain spectral measurements close to harvest day when it is important to determine grape TSS content using reliable and non-destructive techniques.

Few studies have used spectroscopy for the direct measurement of intact and fresh grape berries or leaves [12]. Evaluation of TSS using VIS, NIR, and SWIR spectroscopy is fast, cost-effective, environmentally friendly, non-destructive, reproducible, and repeatable [13]. The objective of this research was to determine the suitability of VIS-NIR-SWIR spectroscopy to estimate TSS in grapes. Spectral reflectance (range of wavelength: 350–2500 nm) data collected from intact and fresh berries were used as predictive variables for PLS regression. Easy-to-use (few factors) predictive models were developed for a few samples, and we selected the best spectral range to avoid complex, overfitted, and unrealistic regression models.

## 2. Materials and Methods

### 2.1. Study Area and Grape Sampling

This work was carried out in vineyards belonging to three DO areas: Bierzo (northwest Spain), Ribera del Duero (north-central Spain), and Rueda (northwest-central Spain). Grapes were sampled in four vineyards with different cultivars: two corresponding to red grapes (Mencía and Tempranillo) and two to white grapes (Godello and Verdejo). Table 1 shows detailed information about sampling.

**Table 1.** General characteristics of sampled vineyards (geographic coordinates referred to WGS84).

| Municipality | Designation of Origin | Grape Cultivar | Longitude | Latitude | Grape Colour |
|---|---|---|---|---|---|
| Camponaraya | Bierzo | Godello | 6.692 W | 42.606 N | White |
| Cacabelos | Bierzo | Mencía | 6.754 W | 42.626 N | Red |
| Valbuena de Duero | Ribera de Duero | Tempranillo | 4.391 W | 41.631 N | Red |
| Matapozuelos | Rueda | Verdejo | 4.765 W | 41.364 N | White |

A total of 12 blocks of 10 vines were defined for each vineyard. Only 5 vines were sampled in each block, yielding 60 samples for each of the 4 grape cultivars, i.e., 240 grape samples in total. A week before harvest, 100 berries were picked from each sampled vine and immediately placed into a sealable plastic bag for storage in an insulated cooler to minimize fruit degradation until the spectroscopic and refractometric measurements. Samples of 100 grapes were used in order to obtain representative reflectance and TSS data from all the sampled vines rather than obtaining data from individual grapes.

### 2.2. Materials and Methods

The proposed procedure was based on using the samples that the winegrower usually takes before carrying out the harvest. The methodology involved data acquisition (grape sampling, TSS, and spectral measurements), spectral data processing (standard normal variate (SNV) transformation and spectral subset organization), and PLS regression (model fit, validation, and selection). Figure 1 summarizes the workflow.

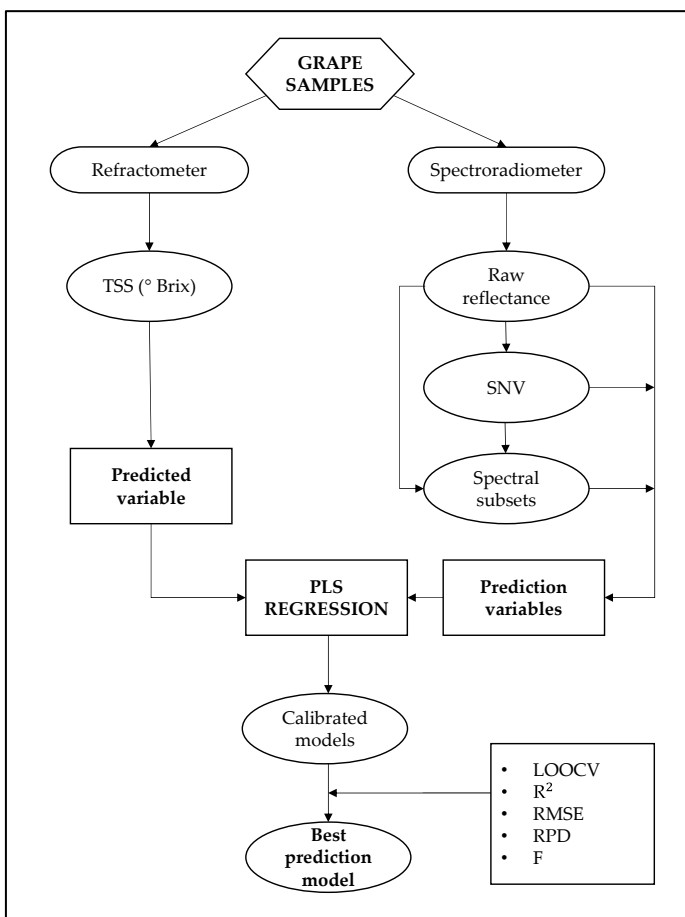

**Figure 1.** Flowchart depicting TSS estimation by spectroscopy. TSS: total soluble solids content; $R^2$: coefficient of determination; SNV: standard normal variate; PLS: partial least squares; LOOCV: leave-one-out cross-validation, RMSE: root mean square error; RPD: ratio of performance to deviation; F: number of PLS regression factors.

### 2.2.1. Spectral Reflectance Measurements

Grape reflectance was measured using a FieldSpec 4 ASD spectroradiometer (Analytical Spectral Devices, Boulder, CO, USA), which detects reflectance in the 350–2500 nm spectral region. The fiber optic cable (25° field-of-view) of the spectroradiometer was coupled with a pistol grip to maintain the cable position at 0.15 m above the grape sample. The light source was a tungsten halogen lamp located in the zenith at 0.50 m over the sample, and the angle between light and fiber was 40°.

Figure 2 illustrates the procedure for obtaining the spectral data. Grape samples (60 samples per grape cultivar with 100 berries each) were spread out in black container cores (17 × 17 cm). Reflectance data were acquired from the 100 berries in 4 different readings (rotating the sample 90° clockwise before each measurement). The reflectance data were then preprocessed to obtain the average of the 4 readings. The sequence of measurements started with a white reference panel (Spectralon 99% reflectance: Labsphere, North Sutton, NH), followed by each reading of the grapes. Recalibration was performed after each measurement of 20 samples.

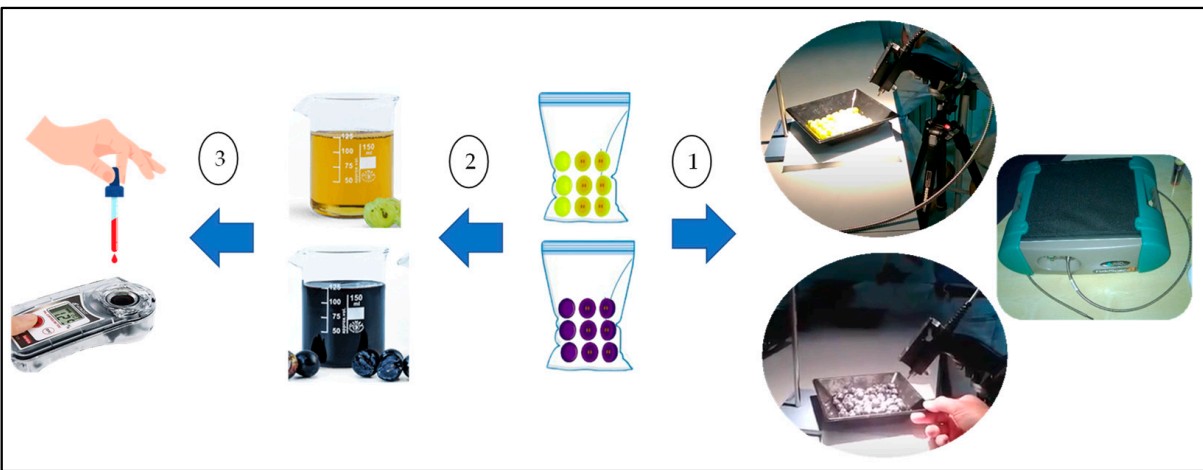

**Figure 2.** Procedure to obtain spectral data using an ASD spectroradiometer and refractometer.

After the spectral data were captured, the grape samples were crushed to obtain the must. TSS (°Brix) was determined at 20 °C using an Atago PR1 digital refractometer (Atago Co., Tokyo, Japan), with a range of 0–45 °Brix and accuracy of ±0.1 °Brix.

### 2.2.2. Spectral Data Preprocessing

Reflectance data were preprocessed using ViewSpect Pro 6.0 (Analytical Spectral Devices, Inc., Boulder, CO, USA) and SAMS 3.2 (Center for Spatial Technologies and Remote Sensing-CSTARS, University of California, Davis CA, USA; https://code.google.com/archive/p/cstars-sams/ Access: 7 November 2022). Outliers in the spectral data measured (4 readings for each grape sample) were identified by a visual analysis (differences in reflectance greater than 0.15 in any range of the spectrum), which were eliminated, and an average spectral signature was calculated per grape sample. A total of 4 spectral subsets were stabilized in order to identify the most suitable wavelength range to estimate TSS: VIS (350–700 nm), NIR (701–1000 nm), SWIR (1001–2500 nm), and the full range (350–2500 nm). Scatter was corrected by SNV transformation, which removed multiplicative interferences of scatter effects from spectral data by centering and scaling each spectral signature [4].

### 2.2.3. TSS Estimation by Spectroscopy

PLS regression was used to estimate TSS (the predicted variable) from the spectral signatures (the predictor variables). PLS is a linear multiple regression procedure that can reduce a large number of collinear spectral signature variables to a smaller number of non-correlated hidden variables or factors [4]. Previous to PLS regression, a principal component analysis (PCA) was carried out in order to identify potential sample outliers: a sample was identified as an outlier in the variance vs. leverage plot (samples with both high values).

Several models were calibrated for each grape variety to identify the most suitable procedure to estimate TSS by VIS-NIR spectroscopy. Two reflectance datasets were considered: non-preprocessed data and SNV-transformed data. An independent model was fitted to each spectral subset (VIS, NIR, SWIR, and the full range) and each grape variety.

An analysis of bandwidth and spectral position was carried out to identify suitable spectral ranges to estimate the TSS. Bands were taken from a very narrow bandwidth of 20 nm and incremented in 5 nm steps (i.e., twice the spectral resolution) up to 2150 nm. A total of 91,378 PLS regression models were analyzed for each dataset. Used as the comparison criterion was the RMSE of each model. The RMSE results were obtained directly from the plsregress Matlab function using cross-fold validation with k = 9.

The regression models were evaluated using leave-one-out cross-validation (LOOCV). $R^2$ and RMSE values were calculated to evaluate model precision and accuracy. Models were compared regarding requirements to fit a robust PLS regression model: small RMSE, high $R^2$, and (as a metric for model complexity [4]) an optimal number of factors (F). The optimum number of factors was selected from the explained variance plot where it is reached when the variance reaches a plateau or peak. The RPD, i.e., the ratio of TSS standard deviation (SD) to standard error (SE), was used to test the usability of the calibrated models. An RPD value of 2 or more was considered appropriate for spectroscopy estimations [14].

### 3. Results

*3.1. Berry Reflectance Spectra*

Figure 3 shows, for the four grape varieties, the range of mean reflectance values and TSS reflectance values for the raw (Figure 3a) and SNV-transformed data (Figure 3b). In Figure 3a, the red grape varieties (Mencía and Tempranillo) had very similar spectral signatures. While reflectance values in the VIS range were similar, some differences were evident from the wavelength at 675 nm, and maximum and minimum reflectance values were found at approximately 895 nm and 1080 nm and at 675 nm and 960 nm, respectively. The spectral signatures of the white grapes (Godello and Verdejo) were different from those of the red grapes but quite similar to each other. Godello and Verdejo presented the highest reflectance values at 570 nm, 830 nm, and 890 nm in the VIS-NIR range. In this range, the reflectance values presented slight differences, although they had the same spectral signatures. From wavelength at 1160 nm, reflectance values for the four varieties were identical.

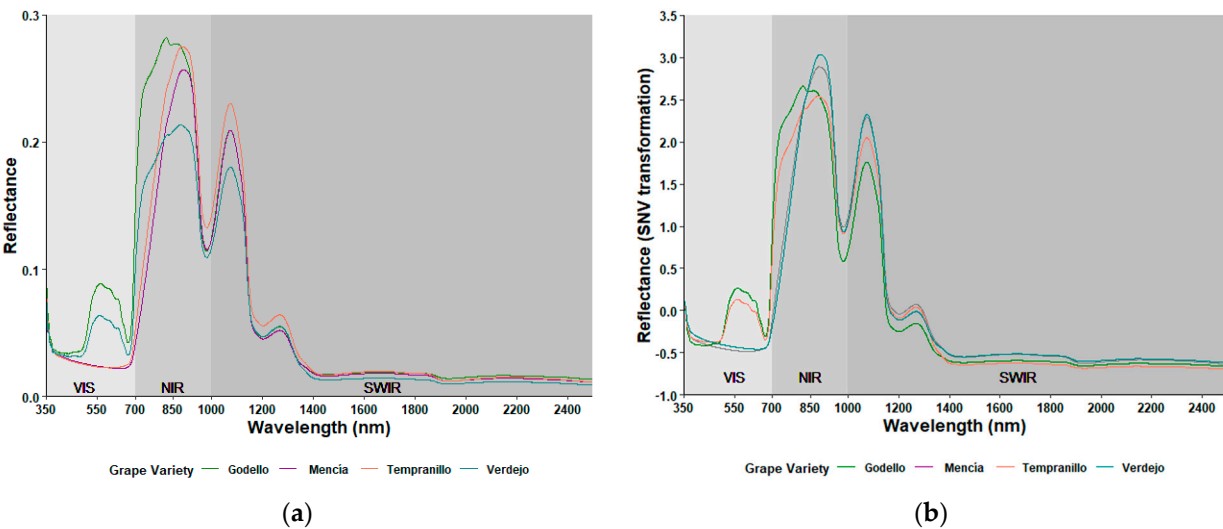

(**a**)  (**b**)

**Figure 3.** Range of the mean spectra of raw (**a**) and SNV-transformed (**b**) values for berries sampled from the four grape varieties (Mencía, Godello, Tempranillo, and Verdejo).

Figure 3b shows that white varieties (Godello y Verdejo) had a notable change in reflectance values, especially in the VIS-NIR range, due to SNV transformation.

### 3.2. Laboratory Analysis

Table 2 shows basic TSS statistics for the berry samples. Although the data come from four varieties located in three different DO areas, the values were quite similar, with small coefficient of variation (CoV) values.

**Table 2.** Descriptive statistics for berry sample TSS (°Brix).

| Varieties | N | Min | Max | Range | Median | Mean | SD | CoV (%) |
|---|---|---|---|---|---|---|---|---|
| Godello | 57 | 17.60 | 25.40 | 7.80 | 22.90 | 22.49 | 1.94 | 8.63 |
| Mencía | 58 | 20.80 | 25.60 | 4.80 | 23.75 | 23.52 | 1.04 | 4.43 |
| Tempranillo | 59 | 16.50 | 24.00 | 7.50 | 21.60 | 21.27 | 1.54 | 7.23 |
| Verdejo | 77 | 17.80 | 23.20 | 5.40 | 20.50 | 20.55 | 1.06 | 5.13 |

TSS: total soluble solids. N: number of samples. Min: minimum value. Max: maximum value. Range: Max–Min. SD: standard deviation. CoV: coefficient of variation. CV = (SD/mean) × 100.

Regarding ripeness level, Mencía had the highest value (minimum 20.80 °Brix and maximum 25.60 °Brix), and Tempranillo had the lowest value (minimum 16.50 °Brix and maximum 24.00 °Brix). The CoV was highest for Godello (8.63%) and lowest for Mencía (4.43%). Sample distribution, in general, was asymmetrical due to the fact that the mean values were lower than the median values, except for Verdejo.

### 3.3. PLS Regression Model Predictions

In the interest of brevity, reported in what follows are only the best results for adjusted PLS models. Supplementary Materials S1 contains detailed information that supports these results (Table S1).

Table 3 shows the $R^2$ values for the PLS regression models using the full spectral range (350–2500 nm) of raw and SNV-transformed spectral signatures. In most cases, $R^2$ and RMSE values were better for the SNV-transformed data than for the raw data. RPD values were acceptable, given that they were in the range of 1.6–2.0 (>2.0 and <1.6 are considered excellent and poor, respectively [15]). Regarding F, the SNV transformation required a similar number to the raw data: slightly fewer for Mencía and increasing from 1 to 7 for Verdejo.

**Table 3.** LOOCV statistics for PLS regression using the full spectral range (VIS + NIR + SWIR: 350–2500 nm) for raw and SNV-transformed data.

| Varieties | | Raw Data | | | | | SNV-Transformed Data | | | |
|---|---|---|---|---|---|---|---|---|---|---|
| | N | $R^2$ | RMSE (°Brix) | SE (°Brix) | RPD | F | $R^2$ | RMSE (°Brix) | SE (°Brix) | RPD | F |
| Godello | 57 | 0.55 | 1.32 | 1.33 | 1.46 | 7 | 0.61 | 1.22 | 1.23 | 1.58 | 7 |
| Mencía | 58 | 0.68 | 0.59 | 0.60 | 1.74 | 5 | 0.68 | 0.60 | 0.60 | 1.72 | 3 |
| Tempranillo | 59 | 0.50 | 1.10 | 1.11 | 1.39 | 7 | 0.64 | 0.94 | 0.94 | 1.63 | 7 |
| Verdejo | 77 | 0.11 | 1.00 | 1.01 | 1.05 | 1 | 0.13 | 0.99 | 1.00 | 1.06 | 7 |

N: number of samples. $R^2$: coefficient of determination. RMSE: root mean square error. SE: standard error. RPD: ratio of performance to deviation. F: number of factors. SNV: standard normal variate.

Table 4 shows the PLS regression models for the NIR spectral range (701–1000 nm). $R^2$ values were generally higher than for the models based on the full range. $R^2$ values for the SNV-transformed data were higher than those for raw data, while RMSE values were lower, except for Verdejo, for which the RMSE value was double. Regarding F, SNV-transformed data required a similar number to the raw data.

Figure 4 depicts variations in weighted regression coefficients with respect to wavelengths. The fact that the range 400–1000 nm contains the most important coefficients in the regressions explains why predictions based on the NIR spectral range (Table 4) proved better than those for the full spectrum (Table 3). The spectral behaviors of the four grape

varieties were very different, while the maximum and minimum weight values were in the range of 400–1500 nm, which is the range that provides information on the chemical groups present in grape skins. Supplementary Materials S2 contains complementary information about weighted regres-sion coefficients of the best models for grape varieties (Figure S1).

**Table 4.** Cross-validation statistics for PLS regression using the NIR spectral range (NIR: 701–1000 nm) for raw and SNV-transformed data.

| Varieties | Raw Data | | | | | | SNV-Transformed Data | | | | |
|---|---|---|---|---|---|---|---|---|---|---|---|
| | N | $R^2$ | RMSE (°Brix) | SE (°Brix) | RPD | F | $R^2$ | RMSE (°Brix) | SE (°Brix) | RPD | F |
| Godello | 57 | 0.75 | 0.98 | 0.99 | 1.97 | 7 | 0.77 | 0.94 | 0.94 | 2.06 | 7 |
| Mencía | 58 | 0.72 | 0.55 | 0.56 | 1.87 | 7 | 0.74 | 0.54 | 0.54 | 1.91 | 7 |
| Tempranillo | 59 | 0.59 | 0.99 | 1.00 | 1.54 | 6 | 0.63 | 0.94 | 0.95 | 1.63 | 6 |
| Verdejo | 77 | 0.38 | 1.12 | 1.12 | 0.94 | 6 | 0.61 | 0.67 | 0.67 | 1.58 | 6 |

N: number of samples. $R^2$: coefficient of determination. RMSE: root mean square error. SE: standard error. RPD: ratio of performance to deviation. F: number of factors. SNV: standard normal variate.

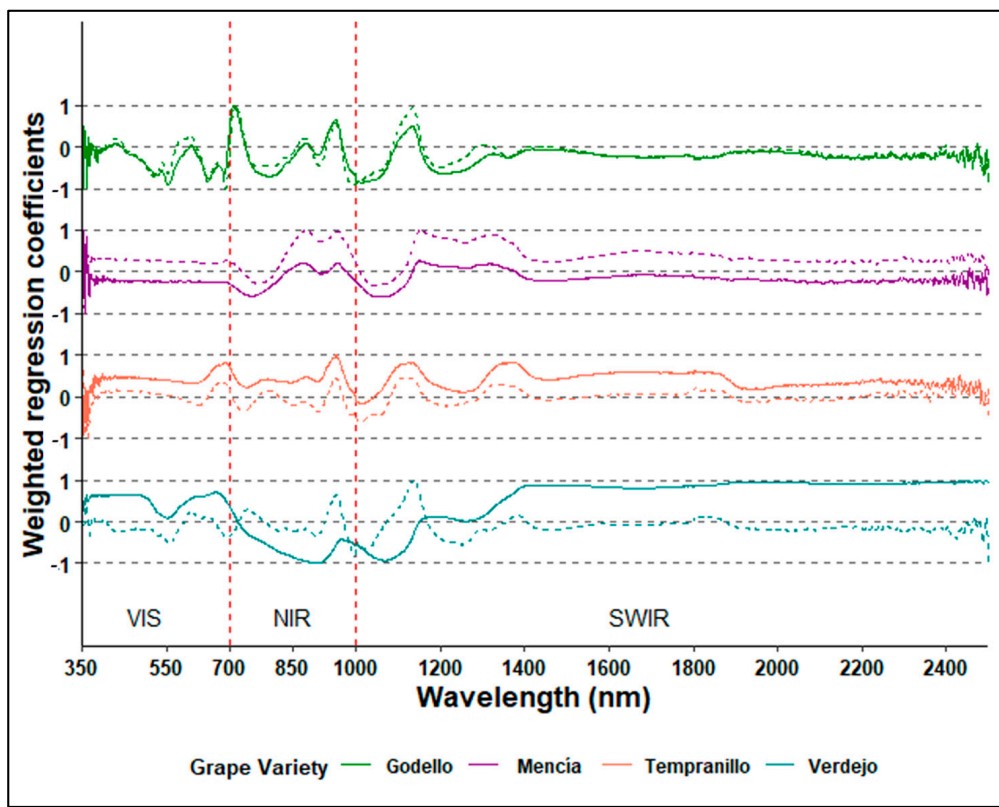

**Figure 4.** Distribution of weighted regression coefficients over the full spectral range for the Godello, Mencía, Tempranillo, and Verdejo grape varieties, as obtained from PLS regressions using raw data (continuous line) and SNV-transformed data (broken line). Must properties for the four varieties were analyzed for cross-validation. Black lines represent zero correlation and have an offset of 3.0 units for clarity of presentation.

For the red grape varieties, the maximum spectral peaks for the raw data and SNV-transformed data were located at 540 nm, 610 nm, and 810 nm for Mencía and at 340 nm, 610 nm, 790 nm, and 1010 nm for Tempranillo, while the minimum peaks were at 700 nm and 1100 nm for Mencía, and at 390 nm, 670 nm, and 900 nm for Tempranillo. In this case, the regression coefficients for the SNV-transformed data were similar to those for the raw data.

As for the white grape varieties, the regression coefficients were somewhat different for the raw data and the SNV-transformed data, especially for Verdejo. For Godello, both regression coefficients showed similar maximum and minimum spectral peaks, located at 370 nm, 605 nm, and 790 nm, and at 340 nm, 670 nm, and 900 nm, respectively. For Verdejo, the spectral peaks for the raw data and the SNV-transformed data were very different in the VIS-NIR range. For the raw data, the maximum and minimum peaks were evident at 500 nm, 680 nm, and 1400 nm, and at 550 nm, 910 nm, and 1080 nm, respectively; however, for the SNV-transformed data, the corresponding peaks occurred at 750 nm, 960 nm, and 1140 nm, and at 560 nm, 700 nm, and 1000 nm. From 1250 nm, the regression coefficients for the SNV-transformed data were very different from those for the raw data.

Figure 5 shows that the reflectance values most correlated with TSS were in the 700–750 nm range, and this correlation was negative. $R^2$ values varied between $R^2 = -0.59$ for Mencía and Tempranillo and $R^2 = 0.38$ for Verdejo. The Mencía variety also had another relative maximum regression value at 1000 nm ($R^2 = 0.45$). The correlogram for Godello was different from the others, with $R^2 = 0.43$ in the range 600–700 nm.

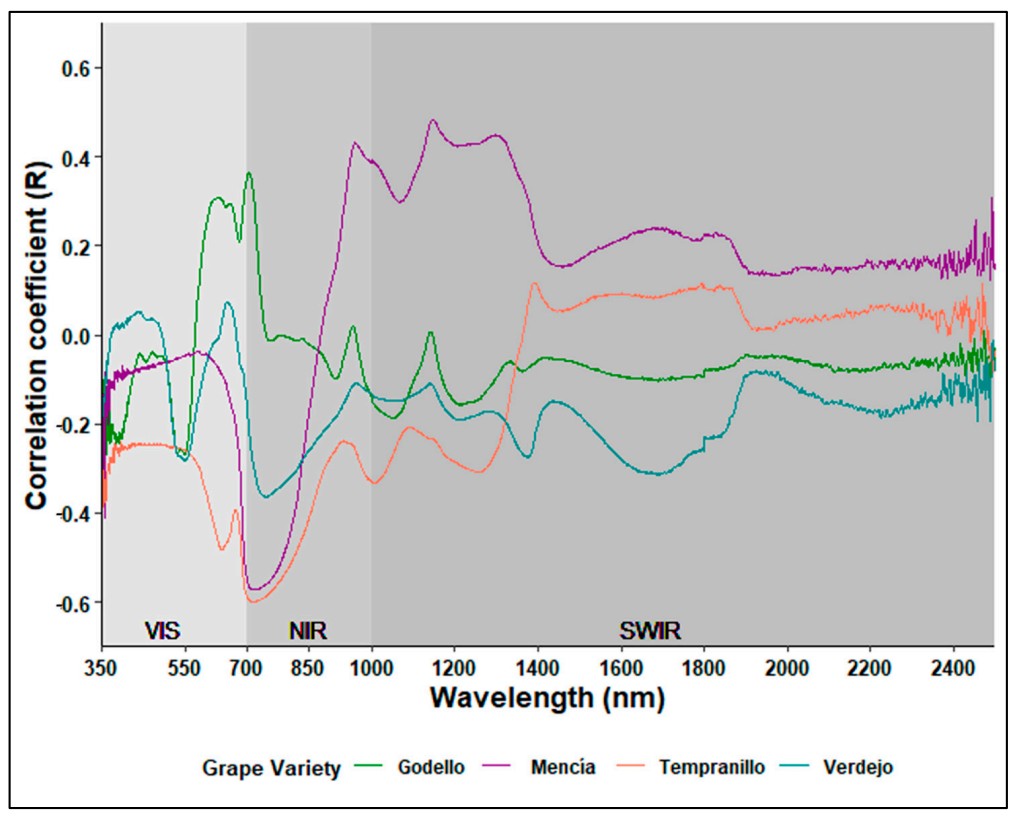

**Figure 5.** TSS correlograms obtained from simple linear correlations with raw spectral reflectance data for each wavelength.

Figure 6 shows the RMSE values for the PLS regression of TSS predictions compared with the corresponding reference data. The bandwidth considered in the model is represented on the vertical axis, while the central wavelength of the band is shown on the horizontal axis. The peak of each triangle represents the PLS regression result when the full spectrum (maximum width) was used for the predictor variables. All other points in the triangle represent RMSE values for different PLS predictions of TSS and spectral bands of varying widths and locations. This comparison enabled the best variables for estimating TSS to be determined and also the influence of the accuracy range of the PLS regression models.

RMSE values were between 0.4 °Brix and 2 °Brix. The best and worst results (lowest and highest RMSE values, respectively) were obtained for the Mencía and Godello varieties.

In all cases, SNV transformation slightly improved RMSE values (Figure 6e–h, and Table 4). Certain areas showed better correlations, which indicated that broad areas of the spectrum are undeniably correlated with TSS—specifically, at around 860 nm for bandwidths of around 201 nm for Godello, at 883 nm for bandwidths of around 232 nm for Mencía, at 916 nm for bandwidths of around 230 nm for Tempranillo, and at 1055 nm for bandwidths of around 230 nm for Verdejo.

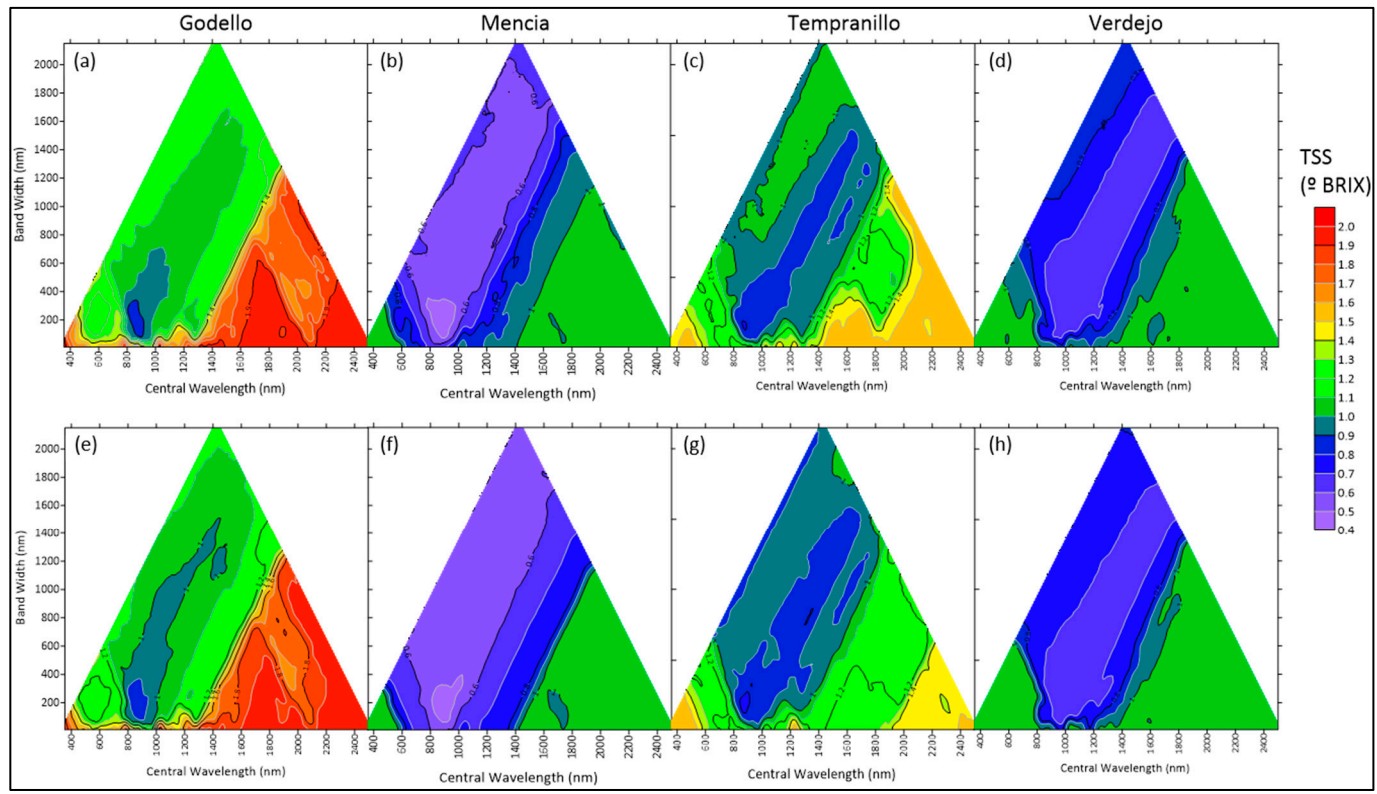

**Figure 6.** RMSE values obtained by PLS regression using raw (**a**–**d**) and SNV-transformed (**e**–**h**) reflectance data. All graphs apply the same color scale (see legend on the right).

Figure 7 shows the observed versus predicted values of the best models to predict the TSS content for each variety. Prediction models of white grapes (Godello and Verdejo) achieved the best results with SNV-transformed data in the NIR spectral range. While red grapes (Mencía and Tempranillo) obtained the best results using SNV-transformed data in the NIR spectral range and SNV-transformed data in the full spectral range, respectively. The better adjustment to the regression line and line 1:1 is presented for Godello (Figure 7a) and Mencía (Figure 7b). The distribution of values is between 17 and 16 °Brix for Godello and 21 and 26 °Brix for Mencía, with the presence of outliers. For Tempranillo (Figure 4c), the distribution of the values is found between 20 and 24 °Brix for the observed and predicted values, with some outliers in TSS values under 19 °Brix. Verdejo (Figure 7d) presents a huge dispersion of the TSS observed and predicted values with respect to the regression and the 1:1 line.

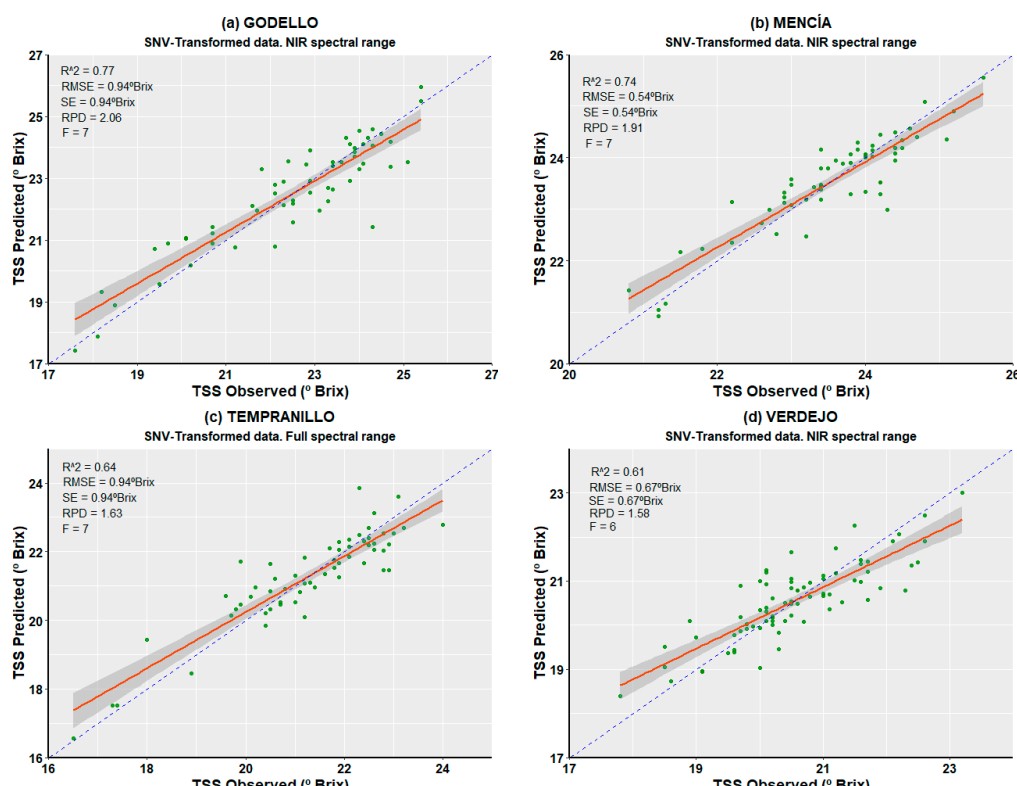

**Figure 7.** Observed versus predicted values of the best models for Godello (**a**), Mencía (**b**), Tempranillo (**c**), and Verdejo (**d**).

## 4. Discussion

The values obtained in Table 2 corroborate those reported elsewhere: a range of t in intact grape berries using contactless VIS and NIR spectroscopy during ripening [8] and a range of 15.6–27.9 °Brix for the Chardonnay variety using refractometry [3]. Our values obtained were in the range of 16.50–25.60 °Brix, considered normal when measured around a week before harvest. Godello was the variety with the highest range (7.80 °Brix), SD (1.94%), and CoV (8.63%). These values indicate the heterogeneity and dispersion of the TSS values. The more homogenous samples were from the Mencía variety, with a CoV of 4.43%.

According to the PLS regression models (Table 3) using full range spectra (VIS + NIR + SWIR: 350–2500 nm), the best results with raw data were obtained for Mencía ($R^2$ = 0.68, RMSE = 0.59 °Brix, RPD = 1.74, and F = 5). However, the PLS regression models using SNV-transformed data improved the statistical values of all the grape varieties except for the $R^2$ and RPD for Mencía and F for Verdejo, which increased.

As for the PLS regression models using the NIR range spectra (Table 4), the best statistics were obtained for Godello: $R^2$ = 0.75, RMSE = 0.98 °Brix, RPD = 1.97, and F = 7. For red grapes, Mencía obtained the best results ($R^2$ = 0.72, RMSE = 0.55 °Brix, RPD = 1.87, and F = 7). SNV transformation yielded better $R^2$ values, reduced RMSE values, and improved RPD values for all the varieties for the same F. Using the lowest possible F in PLS regression models is recommended, as it keeps the models simple [15].

For white grapes measured in laboratory conditions, the $R^2$ = 0.71 and RPD = 1.89 values were lower and higher, respectively [15], than the values obtained for the Godello variety in our research.

In a study of the Cabernet Sauvignon, Chardonnay, and Carménère varieties, for the 650–1100 nm and 750–1100 nm spectral regions and combinations of smoothing, derivative, and path length correction, it was concluded that the spectrum corresponded mainly to a particular zone of the berry (the part closest to the probe) and not to the whole sample, that the °Brix value represented TSS, and furthermore, that the most relevant

information was in the infrared part of the spectrum [3]. The same study obtained $R^2$ values of 0.71–0.98 calibrated with an RMSE range of 0.74–0.94 for a large F (F = 7–20).

Regarding NIR spectroscopy field applications, the best models for the prediction of TSS in intact grapes have been reported for a generic model of different red grape varieties, with values of $R^2$ = 0.91, RMSE = 1.24 °Brix, and RPD = 3.40 for the spectral range 640–1300 nm [8,16]. Results obtained in field conditions were much lower [1] than those obtained in laboratory conditions ($R^2$ 0.38) using a higher F (F = 13).

The use of transformation techniques such as SNV has been reported to significantly improve PLS regression model results [8]. In our research, SNV transformation generally decreased RMSE values and increased $R^2$ and RPD values. This improvement can be attributed to the elimination of deviations caused by particle size and scattering effects in SNV-transformed data [17].

The wavelength region with relevant weighted regression coefficients (Figure 4) for estimating TSS in grapes was located between 580 and 1700 nm, which corroborates Power et al. [18]. The major peaks were around 950–1020 nm, a wavelength range corresponding to O-H and N-H 2nd overtones; they represent alcohols, phenols, and amino functional groups, which are the major constituents of water and sugars such as glucose, fructose, and sucrose in grapes [8]. A study that used the absorption bands in the mid-infrared region reported obtaining peaks corresponding to stretching ($CH_2$), stretching (C=O) ester, stretching (C=O) acid, stretching ($COO^-$), bending ($CH_2$), ring vibration, glycosidic bond (C-O-C), and bending (C-O) in 760–2916 nm spectral range [19]. In our study, the spectra of the different varieties of grapes appeared very similar to the naked eye.

Figure 6 shows that the lowest and highest RMSE values were obtained for Mencía and Godello, respectively. These values support the statistical results, especially for RMSE, achieved for the PLS regression models. Both Mencía raw data and SNV-transformed data had the lowest RMSE values (0.4 °Brix) in the bandwidth around 0–400 nm and in the central wavelength around 900 nm, while Verdejo also achieved low RMSE values (0.5 °Brix) in the bandwidth around 0–600 nm and 800–1200 nm in the central wavelength.

## 5. Conclusions

Diffuse spectroscopy measurements were used to estimate TSS content in four grapes varieties (Godello, Verdejo, Mencía, and Tempranillo) using PLS regression models. Based on the results obtained, the best estimates of TSS content were achieved for red varieties, and especially for Mencía.

The most suitable spectral range for TSS predictions was the NIR range (701–1000 nm). Achieved for this spectral range were the highest $R^2$ and RPD values and also the lowest RMSE and F values. This would indicate that the developed regression models were sufficiently robust to estimate TSS content in grapes. SNV transformation of the data further improved the evaluation metric results of the models in all spectral ranges.

The best variables for estimating TSS (Figure 5) were located in lambda 860 nm, wavelength 201 nm for Godello; lambda 883 nm, wavelength 232 nm for Mencía; lambda 916 nm, wavelength 230 nm for Tempranillo; and lambda 1055 nm, wavelength 230 nm for Verdejo. These optimal points presented the lowest RMSE values.

Our methodology shows that it is possible to estimate TSS content using reflectance values measured by diffuse spectroscopy, a rapid and non-invasive technique for taking field measurements. Furthermore, collecting grape sets representative of vines (instead of individual berries) close to the day of harvest reduces the variability of the samples, which allows us to develop fitted and realistic predictive models.

The next step in this research will be to validate our results by taking measurements directly from grape bunches in the field. This validation will enable the development of a low-cost measurement tool that uses the NIR spectral range to determine TSS content in grapes. Another possible line of continuation of the work consists in the use of hyperspectral images by means of field cameras.

**Supplementary Materials:** The following are available online at https://www.mdpi.com/article/10.3390/agronomy13092275/s1, Figure S1: Distribution of weighted regression coefficients of the best regression models for the Godello (**a**), Mencía (**b**), Tempranillo (**c**), and Verdejo (**d**) grape varieties; Table S1: Cross-validation statistics for Random Forest regression models using VIS (400–700 nm), NIR (701–1000 nm), SWIR (1001–2500 nm), and full spectral range (VIS + NIR + SWIR: 400–2500 nm) for raw and SNV-transformed data.

**Author Contributions:** Conceptualization, K.B.M.-C. and J.R.R.-P.; methodology, J.R.R.-P., V.M. and E.S.-A.; software, K.B.M.-C. and E.S.-A.; formal analysis, K.B.M.-C. and J.R.R.-P.; investigation, V.M., E.S.-A., K.B.M.-C. and J.R.R.-P.; writing—original draft preparation, K.B.M.-C., V.M. and J.R.R.-P.; writing—review and editing, K.B.M.-C. and J.R.R.-P.; supervision and project administration, J.R.R.-P. All authors have read and agreed to the published version of the manuscript.

**Funding:** This research received no external funding.

**Data Availability Statement:** Data sharing is not applicable.

**Acknowledgments:** The authors acknowledge the assistance of María T. Alonso-Rodríguez and Ana I. Barge-Carrete for providing support in the lab analysis.

**Conflicts of Interest:** The authors declare no conflict of interest. The authors declare that they have no known competing financial interests or personal relationships that could have influenced the work reported in this paper.

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
