# Peer review of "Total Soluble Solids in Grape Must Estimation Using VIS-NIR-SWIR Reflectance Measured in Fresh Berries"

_agronomy, doi:10.3390/agronomy13092275_

Round 1

Reviewer 1 Report (Previous Reviewer 2)

The paper presents a methodology using PLSR to predict TSS from four grape varieties. The novelty is on the lower side, as this has been demonstrated before. The authors needs to improve the description on some parts of their methodology before this paper is accepted. Particularly noteworthy is the fact that the PLSR models developed from the full spectral range are (significantly) inferior to the ones developed only using NIR, something that in my humble opinion needs to be supported using an additional ML algorithm.

Major remarks:

* It should be clearly stated why you opted to use the pistol grip and not come into direct contact with the berries themselves (as some works do) but use the approach customarily employed when using an optical lens or a hyperspectral camera. Your set-up e.g. makes sense for me to be used in a dark box with the lens of the instrument.

* Line 128 to 130: please report more rigorously what the outlier analysis entailed

* The caption of Figure 3: this is not the range (extent) but rather the mean spectra?

* It's not clear how LOOCV was performed, particularly because you report integer values for F in Tables 3 and 4. By that I mean is that in LOOCV you have N models (one for each sample of your dataset). Each of these is trained in the N-1 data, where a grid search takes place to optimize the number of components and a model is fit which is then used to predict the left-out datum. As such, it's possible to have models with different number of latent variables. This is not clearly described in your manuscript and you may have erroneously calculated some of your results.

* It's particularly interesting to note that the complete spectrum yields worse results. Please test another ML algorithm as well, e.g. the Random Forest, to ascertain whether this is due to PLS's poor generalization performance. Adding more features should not cause that a significant drop in performance in apt ML models.

* Figure 4 is from the models that used the entire spectrum, correct? This is not clearly stated. It'd be interesting to see the comparison with the NIR models particularly for Verdejo + SNV.

* Figure 5: The y-axis is not Reflectance

* Lines 220-239: The discussion is jointly for Figures 4 and 5 but this is not clear to the reader

* A scatter plot depicting the real vs predicted values of the best models for each variety would aid the reader

* There are only a few citations in the manuscript, particularly for a journal paper this is noteworthy.

Minor remarks

* Line 106: CO, USA?

* Line 108: Did you do this with an angle as noted in Figure 2? Please describe the geometry incl. the angle between light and fiber.

* Line 114: pre-processed

A few mistakes are noted in the minor remarks

Author Response

Reviewer 2 Report (Previous Reviewer 1)

1. Why authors directly  used SNV as the unique pre-processing technique and no other one, considering that, usually, many different techniques are investigated (such as SNV, MSC, derivatives, and their different combinations...)?

2.  What is about the training samples and validation samples?

3.  In section 3.1.  authors should describe which the reflectance peaks are associated with molecular bonds and characteristic functional groups.

Author Response

Reviewer 3 Report (New Reviewer)

Dear Authors,

General Comment. The work is of scientific relevance and is well-written. However, some considerations need to be pointed out.

Introduction - The narrative in the introduction is well-composed. However, the objective of the work conveys the interpretation that the model was applied to individual intact grapes. Yet, when one goes to the materials and methods section, it becomes evident that the spectrum was collected from a set of grape berries and not individually.

Materials and Methods - The text described between lines 110 to 121, associated with Figure 2, indicates that the spectra were collected from a sample of 100 grape berries, not individually, meaning 1 or 2 spectra per berry. Consequently, the content of soluble solids was obtained from a group of fruits. We understand that NIRS techniques can be used in a production line in the grape juice and wine industry. However, to achieve better reliability and robustness, the model should be built fruit by fruit in the case of grapes, thus minimizing the prediction error of the model. This is because each berry may have differences in the content of soluble solids, and averaging these values may mask this information. The collection and measurement of TSS should be done as shown in Urraca et al. (2016), a reference cited in the manuscript. A single grape berry provides enough juice to perform soluble solids measurements, which, in our view, does not justify collecting spectral data and quantifying the response variable in a sample of 100 berries. The authors need to justify this methodology. Also, the analysis of principal components is missing as a tool for evaluating spectral behavior and potential outliers. It is not clear how the removal of outliers from the sample was performed. The spectra were centered on the mean, but further details are needed.

Results and Discussion - The results were influenced by how the spectra were collected. It is necessary to present the spectra after the SNV pre-processing step. The reason for the low R² values of the PLSR models may be due to the methodology of spectral data collection and subsequent quantification of the response variable (TSS). The authors mentioned that they would run the model in four spectral ranges: VIS, NIR, and SWIR, but only the results for the full spectral range (Table 3) and NIR (Table 4) were presented.

Conclusion - The conclusion should directly address the objective of the work and highlight the advancements achieved. As it is currently written, it seems more like a discussion of the results rather than a clear and direct response to the research objective.

Our advice is that the work should be reconsidered for a new revision, provided that the authors can address and fix the points mentioned.

Round 2

Reviewer 1 Report (Previous Reviewer 2)

The authors have done a well-rounded effort to reply to all of my points. I am afraid that I was perhaps not 100% clear on some of them though, as some of the replies are not entirely what I was expecting / asked. I hope that my comments below are more understandable by the authors.

Point 1.0: From your reply "The novelty of the work described in the manuscript is that it measures reflectance directly on the grape berries (not on the must or bunches)". This is not 100% novel (never done before anywhere in the world) as many studies that employ a point spectrometer have been used on grape berries (e.g., https://doi.org/10.1255/jnirs.566, https://doi.org/10.1109/TIM.2007.910098, for some old ones). The second part of your reply is perhaps the true novelty. Please re-phrase it in the manuscript.

Point 1.1: Please elaborate a bit more; for example, although I am not sure this is the reasoning: A single berry is not representative of the whole bunch. We also don't want to measure must by crunching them. Measuring many berries simultaneously is more representative. All these with a question mark from my side, of course; but this type of reasoning needs to be in the paper!

Point 1.4: Let me clarify. The optimal number of factors needs to be repeated N times. It's not one single number. If you are doing leave-one-out CV, you use N-1 samples to build a model -> this needs to optimize its factors. To do so, perhaps you use 5-fold internal cross-validation or a resampling strategy. The R package caret has this image: https://topepo.github.io/caret/premade/TrainAlgo.png which describes this process within each fold. Now, when you move to the next out-of-the-bag sample (different N-1 set) you repeat the process. Thus, the image I sent needs to be repeated N times. This is usually referred to as nested cross-validation: https://scikit-learn.org/stable/auto_examples/model_selection/plot_nested_cross_validation_iris.html This is the proper way to perform leave-one-out CV when simultaneously optimizing a hyperparameter.

Point 1.5: This in my opinion needs to be included in the manuscript because it solidifies your remark that indeed a subset is better than the full range as this is demonstrated in multiple ML models. I would advise placing it even in the appendix and adding the necessary discussion.

Point 1.6: Similarly, I would recommend including the coefficients of the best models for each variety in the manuscript - after all, the ones with NIR perform better than the full range, thus it's interesting to see where these ones focus.

Finally, two notes: a) please proof-read your manuscript, e.g. newly inserted lines 366 to 368 are not grammaticaly correct. b) I fail to see any mention about hyperspectral imaging (instead of point spectrometery) as a future step / research avenue

Please proof-read!

Author Response

Reviewer 2 Report (Previous Reviewer 1)

 I checked the manuscript and the article can be accepted.

checked the manuscript and the article can be accepted.

This manuscript is a resubmission of an earlier submission. The following is a list of the peer review reports and author responses from that submission.

Round 1

Reviewer 1 Report

This study used VIS-NIR reflectance to measure total soluble solids in grape. The novelty in this study is poor. 

Reviewer 2 Report

I have carefully read and reviewed the paper which focuses on predicting TSS from VIS-NIR reflectance spectra. The major issue is that there is a lack of novelty in the paper, as it has been demonstrated in the literature that it is possible to estimate TSS using reflectance spectroscopy. Please find my remark below:

1. The introduction needs to improve: i) include more up to date research that has demonstrated the suitability of spectroscopy to estimate TSS (e.g., https://doi.org/10.3390/s23031065, https://doi.org/10.3390/horticulturae8070613, https://doi.org/10.3390/agronomy12092113), for example your 12-16 studies are 15 years old, ii) the novelty of the study needs to come forward and you should highlight what really is new in this research compared to the more recent studies.

2. You may want to include SWIR in the title, as you have defined it at 1000 to 2500 which you include in your study.

3. If I understood correctly from Lines 75 to 78 you performed sampling only on one date; why did you do that and not cover a larger part of the variance of the Brix content? This sampling procedure led to low variance (Table 2) and you have only (if at all) few samples that are under- or over-mature. This needs to be justified.

4. Please include the resolution etc. of the refractometer. Line 103-104 repetition of 79-80.

5. Figure 1 regresion => regression

6. Section 2.2.1: If you used a pistol grip with a fibre optic cable, why did you not come into contact with the grape itself? Please include a photo of the measurement's geometry to understand exactly how each measurement took place.

7. Line 98: four readings, did you take the average after that?

8. Line 100: follow => followed

9. I didn't 100% understand how you collected the data. You have 5 vine trees times 12 blocks = 60 vine trees per cultivar, in total 240 sampled trees (you mention "240 grape samples" which confuses me). If you picked 100 berries from each sampled vine, this means that you have 24000 individual berries? (Lines 75 to 77). When you measure these spectrally, did you record 24000 spectral signatures and measure the Brix in each one? Did you place each single berry in the black container core? (Line 97). But then at Table 2 you only show about 60 berry samples per variety (except for Verdejo that has 77). The way I try to make sense of it is that you placed the 100 berries in the black container core and measured them altogether, thus you had in fact one "data point" (i.e., spectral signature coupled with Brix value) from the 100 berries that each was from a single vine. But this is not clearly described. If this is the case, you lose the information of each individual berry and you group them altogether; thus each "data point" is an amalgation of 100 berries both spectrally and in terms of the must. It should be clear that each bunch of grapes may contain berries of different maturity degrees, and doing so one cannot differentiate between berries. Is this the correct approach?

10. Line 109-110 here you calculate the average (of the four readings?) and in line 112 you average again?

11. Section 2.2.3: it's not clear how you optimized the number of LVs (your F) for PLSR. When performing LOOCV did you perform an internal k-fold CV? What criterion did you use to select the optimal value?

12. Personally, I wouldn't call Godello and Verdejo "green grapes" but "white wine grape varieties" to differentiate from the red grape varieties, but I am not the expert at this domain.

13. Line 161: What does this reference [21] say about what is appropriate for spectrocsopic calibrations? Kindly elaborate.

14. Line 172: 250 => 350

15. Line 174: <2.0 => >2.0

16. Tables 3 and 4 you don't need two digit precision for N

17. Figure 3: Did you think about plotting the absolute values here? To be able to identify the most important regions and not having to discuss about 'maxima' and 'minima'. The x and y axes need labels.

18. Line 213: For Verdejo ...

19. Figure 5: I guess raw is a-d and SNV e-h? Please update the caption.

20. Line 113 and 126 you mention different spectral subsets (VIS, NIR, SWIR and VIS-NIR-SWIR) but in the results I only saw NIR and VIS-NIR-SWIR?

21. I read at lines 127-128 "before calibrating .. the TSS" that before calibrating the final models you select the optimal spectral subsets; but I am not sure if the results (Table 3 and Table 4) which depict the results in their full respective ranges are from the best subset or from the entire range. Similarly, in Figure 5 - do the best resuls from here correspond to either of the tables?

22. The overall quality of the figures (except Figure 5) feels low, you should consider revising them / plotting with another software library or package.

Only minor edits are required, please make sure that both spell and grammar check is on.